# Developing and Field Testing Path Planning for Robotic Aquaculture Water Quality Monitoring

Anthony Davis [1,*], Paul S. Wills [1], James E. Garvey [2], William Fairman [1], Md Arshadul Karim [1] and Bing Ouyang [1,*]

1   Harbor Branch Oceanic Institute, Florida Atlantic University, 5600 US 1 North, Fort Pierce, FL 34946, USA
2   Center for Fisheries, Aquaculture, and Aquatic Sciences, Southern Illinois University, Carbondale, IL 62901, USA
*   Correspondence: anthonydavis2020@fau.edu (A.D.); bouyang@fau.edu (B.O.)

**Abstract:** Marine food chains are highly stressed by aggressive fishing practices and environmental damage. Aquaculture has increasingly become a source of seafood which spares the deleterious impact on wild fisheries. However, continually monitoring water quality to successfully grow and harvest fish is labor intensive. The Hybrid Aerial Underwater Robotic System (HAUCS) is an Internet of Things (IoT) framework for aquaculture farms to relieve the farm operators of one of the most labor-intensive and time-consuming farm operations: water quality monitoring. To this end, HAUCS employs a swarm of unmanned aerial vehicles (UAVs) or drones integrated with underwater measurement devices to collect the in situ water quality data from aquaculture ponds. A critical aspect in HAUCS is to develop an effective path planning algorithm to be able to sample all the ponds on the farm with minimal resources (i.e., the number of UAVs and the power consumption of each UAV). Three methods of path planning for the UAVs are tested, a Graph Attention Model (GAM), the Google Linear Optimization Package (GLOP) and our proposed solution, the HAUCS Path Planning Algorithm (HPP). The designs of these path planning algorithms are discussed, and a simulator is developed to evaluate these methods' performance. The algorithms are also experimentally validated at Southern Illinois University's Aquaculture Research Center to demonstrate the feasibility of HAUCS. Based on the simulations and experimental studies, HPP is particularly suited for large farms, while GLOP or GAM is more suited to small or medium-sized farms.

**Keywords:** aquaculture; robotics; monitoring; machine learning; path planning; vehicle routing problem; drones

## 1. Introduction

Global fish stocks in every ocean have been declining for decades, primarily due to overfishing [1,2]. Changes in the marine environment from human pollution have also brought about further reductions in fish populations and the bioaccumulation of pollutants such as heavy metals and microplastics in wild fish [3,4]. Since 1990, fish consumption has doubled, while capture fishery production has remained constant. The rising demand has been met by the explosive growth of aquaculture, which has grown exponentially over the last few decades, producing more fish than capture fisheries for the first time in 2012 [5].

Aquaculture is a labor-intensive activity, with the most burdensome task being maintaining appropriate water quality in the water body. Of particular importance is the level of dissolved oxygen (DO), which, if mismanaged, will result in sudden collapse of a fish pond. Fish farmers typically measure DO by attaching a sensor to a vehicle which is manually driven to each pond. This process is repeated multiple times a day and is particularly important to measure at night, when DO depletion typically occurs due to the lack of photosynthesis and subsequent plant respiration. Because DO measurement is a repetitive and relatively simple task, it is a prime target for automation.

### 1.1. Previous Work on DO Monitoring

Much advancement has been made toward the use of AI and drones to help manage production and maintenance on aquaculture farms. Besides DO monitoring, methods have been proposed to automate activities such as fish counting, fish length estimation, and facility surveillance [6,7]. Two distinct approaches have been used to tackle the problem of automated DO monitoring. One approach employs DO sensors mounted on the pond buoy, powered by a solar panel and connected to the central office via wireless communications. Such a sensor pod is installed in each pond to constantly monitor DO [8–10]. While this type of solution does reduce labor costs, equipment, and installation costs do not scale well for commercial-sized farms which have thousands of ponds. Additionally, sensors which remain in water will biofoul if they are not regularly maintained. These hardware and maintenance requirements will detract from the goal of reducing overall costs. Another issue is that it provides limited spatial coverage of the pond where the DO level may vary extensively in different locations, especially in a large pond.

The other type of approach to automatically monitor DO levels involves the design of sensor packages [11,12] and vehicles [13,14] to travel between ponds on a regular basis to collect their DO data. This reduces the maintenance burden and eliminates the need to install sensors for each pond. It also enables the measurement of DO levels at different locations in the pond, providing more reliable data. Remote sensing techniques have also been proposed to predict DO levels using multispectral images of the pond surface [15]. While remote sensing would greatly simplify the task for the drones, it introduces the risk of poorly predicting the DO due to the unknown level of oxygenation beneath the surface. Remote sensing, in particular satellite remote sensing, is subject to the weather conditions (i.e., cloud coverage). On the farm, the catastrophic fish loss may occur within an hour if DO depletion is not detected and mitigated in time.

While studies of autonomous aquaculture systems are relatively common, work regarding robotic motion planning for autonomous aquaculture management has been sparse. Motion planning for oyster farm maintenance [16], offshore farm net pen cleaning [17], and mesh networks [18] are some of the few examples. The rising demand for aquaculture products, the short supply of laborers, and so far limited research interest makes this a prime topic for further investigation.

### 1.2. Previous Work on Drone Swarming and Path Planning

Routing drones safely and efficiently is a complex problem. Multi-robot navigation is a well-studied topic of research, with many constraints and requirements [19,20]. Devising the path for each drone is an example of the Vehicle Routing Problem (VRP), a generalization of the Traveling Salesman Problem (TSP). In the VRP, a group of vehicles must visit each node of a graph with the minimal longest route. This ensures the evenest distribution of distance required to be traveled by each drone, which allows the fewest possible drones needed to adequately monitor the farm. Considerable research has been invested in TSP, VRP and its numerous variants [21–23], and as a result many solutions to this routing problem are available. More specifically, this environment is an example of Distance Constrained VRP (DVRP) due to the limiting factor being the range of the drones.

However, most VRP research consists of highly idealized node graphs. While most studies focus on randomly generated node locations with 100 nodes or fewer, the reality of aquaculture farms is commercial farms typical have hundreds of ponds, and may have more than a thousand. Aquaculture farms also dig their ponds in a regular, gridlike manner, which would facilitate repetitive paths. There are relatively few VRP methods which are intended for calculating routes for more than 100 nodes [24,25], and fewer methods for calculating paths in a grid space [26,27]. Another factor to consider is the drones will be required to fly in inclement weather. The impact of wind on their path planning may be highly significant, yet there is little research regarding its impact on drone path planning [28,29]. These unique conditions require special considerations that are not sufficiently studied in the literature.

### 1.3. The Hybrid Aerial Underwater Robotic System

Our proposed solution is the Hybrid Aerial Underwater Robotic System (HAUCS) [30,31]. In this system, DO sensors and radio modules are installed on a set of GPS-navigated drones which take in situ measurements of different ponds on the farm before returning to be recharged. The drones conduct a direct, full depth measurement of DO, temperature, and pressure by submersing the sensor to a depth of up to several meters below the surface to gain a full understanding of the state of each pond. A diagram of the HAUCS concept is shown in Figure 1.

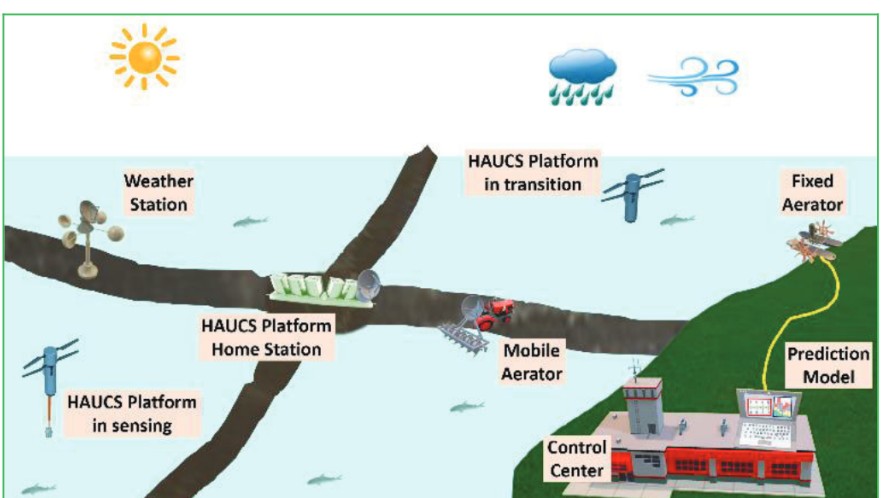

**Figure 1.** Diagram of the HAUCS concept. Data from the HAUCS platforms and weather stations are integrated in a control center, and conditions can be predicted to enable efficient aeration and pond maintenance. Adapted from [31].

## 2. Overview of the Three Path Planning Algorithms

Our previous work investigated three different VRP approaches for this problem using these simulated fish farms [32]. These approaches were the Google Linear Optimization Package (GLOP) [33], Graph Attention Model (GAM) [34], and our heuristic solution, the HAUCS Path Planning algorithm (HPP). These methods were selected because they represent high performing methods with different approaches.

### 2.1. Google Linear Optimization Package

GLOP is a linear optimization Python package that computes the optimal or nearly optimal solution for a set of node locations given certain linear relationships [33]. Linear programming is a method used to optimize a linear objective function, subject to a set of linear inequality or equality constraints. In the context of the vehicle routing problem, GLOP can be used to minimize the maximum route length for a given set of node coordinates, which are converted to a distance matrix. It does this by using a Primal Simplex algorithm, which works by starting with a feasible solution to the problem, called the primal solution. The algorithm then iteratively improves the primal solution until an optimal solution is found. At each iteration, the algorithm chooses a direction to move in, called a pivot, that improves the objective function. The pivot is chosen based on the values of the objective function and the constraints at the current iteration. In this case, the objective is to obtain the minimum longest route. One potential disadvantage of using GLOP for the vehicle routing problem is that the complexity of the algorithm grows exponentially with the size of the problem, making it inefficient for large-scale problems.

### 2.2. Graph Attention Model

Kool et al. proposed a machine learning approach to solving routing problems using a graph attention model (GAM) and reinforcement learning [34]. They use GAM to encode

a learned representation of the locations and their relationships, which is then used in conjunction with reinforcement learning to identify optimal routes. GAM is a type of graph neural network that uses attention mechanisms to weigh the importance of different parts of the input data, allowing it to focus on the most relevant information when making predictions. To train the GAM, Kool uses a dataset of synthetic routing problems with known optimal solutions, and uses reinforcement learning to learn a policy for selecting actions that lead to the optimal routes. GAM is trained using a variant of the actor–critic algorithm, which involves learning both a policy for selecting actions and a value function that estimates the expected future reward. In our experiments, GAM was built with a 128 dimension embedding and 3 encoder layers, with the same dimensions for the actor–critic reinforcement learning trainer. The results of the experiments in the paper show that GAM is able to learn the highly optimal routes for the synthetic routing problems, and outperforms many traditional optimization methods in terms of both accuracy and speed.

### 2.3. The HAUCS Path Planning Algorithm

HPP is a path-planning strategy that enables drone platforms to monitor aquaculture farms in a back-and-forth pattern using a geometric approach. It consists of two phases: clustering and routing. In the clustering phase, k-means is used on the set of nodes to produce evenly spaced clusters, with k set to the number of drones to be used. Convex polygons are then created around each cluster, and the antipodal pairs of these polygons are calculated using the Shamos algorithm [35]. In the routing phase, the Optimal Coverage Algorithm (OCA) is used to determine the shortest optimal back-and-forth path that starts and ends at one of the antipodal point pairs [36]. The route is then assigned to follow the optimal back-and-forth pattern as closely as possible. If a cluster is too small to form a polygon, it is expanded by adding the closest nodes from neighboring clusters until all clusters have a valid size and shape. The performance of HPP is demonstrated by the evenly spaced clusters of approximately equal size shown in Figure 2, despite the irregular outline shape of the node locations. The effectiveness of this approach is dependent on the assumption that the distribution of nodes is generally even. If nodes are placed randomly or have areas of higher density, this assumption may not hold, and the method may be less effective in producing efficient routes. Further details of the HPP implementation can be found in our previous works [32,36,37].

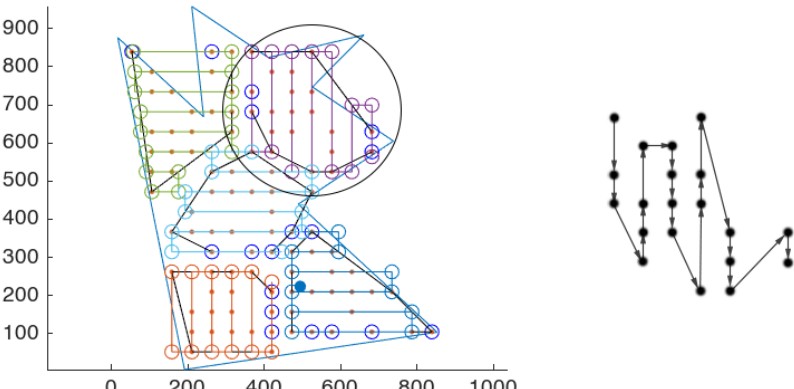

**Figure 2.** Plot of HPP clusters and their associated route plans. The blue dot signifies the depot location. The circled section is shown on the right with the resulting path for that drone.

### 2.4. Simulator Results

To create a realistic distribution of fish ponds, an algorithm was devised using some basic assumptions of how a typical fish farm is organized. First, a randomly generated convex polygon is created to replicate the farm's available land and a grid of points is filled inside it. Then, 20% of the points are removed in order to replicate empty ponds and add additional variation. This creates a pattern of points that are not completely regular yet not

random, and is a reasonably accurate generalization of the spatial distribution of typical fish farms. Examples of a commercial aquaculture farm and our simulated farm are shown in Figure 3.

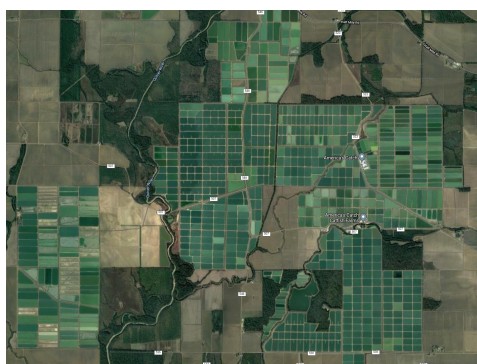 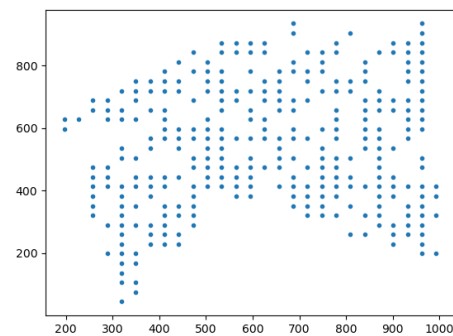

**Figure 3.** (**left**) America's Catch Catfish Farm with approximately 700 ponds. (**right**) Example of 300 pond simulated aquaculture distribution. Adapted from [32].

These three methods are tested on a set of these generated aquaculture farm locations, varying in size from 50 ponds to 700 ponds. The results for these three methods differed depending on the scale of the simulated farm, as shown in Tables 1–3. All results were run in Ubuntu 21.04 with an AMD 5600x CPU, NVIDIA RTX 3080 GPU, and 16 GB of 2133 MHz DDR4 RAM. GAM is implemented in Python PyTorch using CUDA to leverage GPU parallelization, GLOP is primarily C++ and Python, and HPP is purely MATLAB. For small farms, which consist of 50 to 200 ponds, GLOP was the most efficient router with the lowest average maximum route length. For larger farms, which were simulated as 200 ponds to 700 ponds, HPP provided the best routing. GLOP is not able to compute routes for farms that large due to the high computational complexity of the method, so the optimal solutions cannot be found in a reasonable amount of time. While GAM was unable to outperform GLOP in small farms or HPP in large farms, it achieved comparable quality of results for large and small farms in a fraction of the time.

**Table 1.** Average Total Cost.

| Ponds | HPP | GAM50 | GAM100 | GLOP |
| --- | --- | --- | --- | --- |
| 50 | 7.62 | 7.19 | 7.6 | **6.37** |
| 100 | 9.94 | 9.45 | 9.85 | **7.72** |
| 200 | 12.66 | 13.06 | 12.95 | **9.12** |
| 300 | **15.37** | 17.22 | 16.47 | - |
| 500 | **19.02** | 25.46 | 22.79 | - |
| 700 | **21.72** | 35.35 | 29.74 | - |

**Table 2.** Average Maximum Route Cost.

| Ponds | HPP | GAM50 | GAM100 | GLOP |
| --- | --- | --- | --- | --- |
| 50 | 2.01 | 1.75 | 1.82 | **1.47** |
| 100 | 2.53 | 2.27 | 2.39 | **1.65** |
| 200 | 3.14 | 2.73 | 3.27 | **1.9** |
| 300 | **3.79** | 4.02 | 3.89 | - |
| 500 | **4.58** | 5.57 | 6.32 | - |
| 700 | **5.28** | 12.77 | 7.66 | - |

**Table 3.** Average Run Time (ms).

| Ponds | HPP | GAM50 | GAM100 | GLOP |
|---|---|---|---|---|
| 50 | **0.013** | 1.1 | 1.2 | 735 |
| 100 | **0.030** | 3.5 | 3.1 | 6581.5 |
| 200 | **0.047** | 9.0 | 10.7 | 54,352.6 |
| 300 | **0.095** | 21.5 | 23.9 | - |
| 500 | **0.110** | 59.7 | 64.9 | - |
| 700 | **0.115** | 179.7 | 185.1 | - |

### 2.5. Wind Considerations

Whenever planning flights for any aircraft, it is important to take weather conditions into consideration. Significant wind can dramatically change the optimal flight path by increasing the cost of flying upwind. Some research has been done to investigate the effect of wind on path planning for aerial drones, but this research is divided into fixed wing [28,29] and multicopter [38,39] configurations. The so-called Windy Routing Problem (WRP) [40–42] addresses multicopter flight planning in windy conditions, as it is not constrained by the turn radius of a fixed wing aircraft, which requires additional consideration [28]. WRP assumes steady wind in a single direction and no complex wind phenomena such as turbulence and pressure or density changes.

To model this for our aquaculture dataset, we increase the cost of travel in upwind directions and reduce the cost in downwind directions. A random direction and wind speed is chosen, with the maximum wind speed being just less than the top speed of the drone. This process is described in Figure 4. We repeat the aquaculture simulation path planning with this added factor and report the results in Tables 4–6. GAM did not perform as stably when wind conditions were factored in, and much of its increase is attributed to its total failure to plan efficient routes for a few samples while performing reasonably well for most samples. HPP and GLOP performed similarly as in non-windy conditions, with GLOP providing the optimal routes but being unable to produce results in a reasonable amount of time for real-time operations.

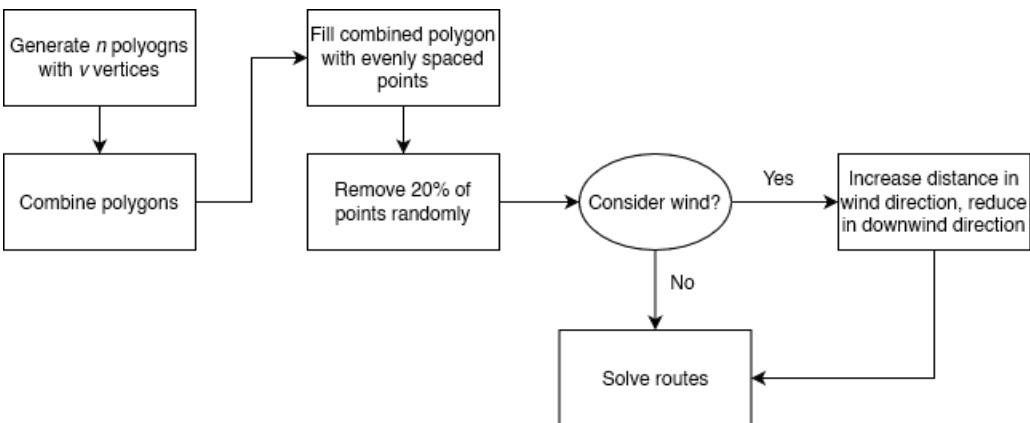

**Figure 4.** Flowchart describing the process of generating the simulated data with and without wind considerations.

**Table 4.** Average Total Cost in Wind.

| Ponds | HPP | GAM50 | GAM100 | GLOP |
|-------|-----|-------|--------|------|
| 50 | 8.69 | 9.28 | 10.69 | **7.017** |
| 100 | 10.41 | 11.65 | 14.79 | **7.71** |
| 200 | 13.99 | 16.99 | 24.02 | **8.95** |
| 300 | **15.02** | 21.31 | 35.95 | - |
| 500 | **9.10** | 32.16 | 63.57 | - |
| 700 | **22.93** | 41.57 | 84.75 | - |

**Table 5.** Average Maximum Route Cost in Wind.

| Ponds | HPP | GAM50 | GAM100 | GLOP |
|-------|-----|-------|--------|------|
| 50 | 2.26 | 2.28 | 3.14 | **1.65** |
| 100 | 2.71 | 2.89 | 6.18 | **1.72** |
| 200 | 3.45 | 6.63 | 10.29 | **1.88** |
| 300 | **3.75** | 7.99 | 9.34 | - |
| 500 | **4.66** | 13.21 | 15.88 | - |
| 700 | **5.64** | 16.89 | 17.21 | - |

**Table 6.** Average Run Time in Wind (ms).

| Ponds | HPP | GAM50 | GAM100 | GLOP |
|-------|-----|-------|--------|------|
| 50 | **0.03** | 0.94 | 1.03 | 748.79 |
| 100 | **0.04** | 2.96 | 3.31 | 6091.04 |
| 200 | **0.06** | 9.10 | 10.67 | 58,157.81 |
| 300 | **0.09** | 22.31 | 28.76 | - |
| 500 | **0.21** | 76.28 | 72.36 | - |
| 700 | **0.21** | 151.22 | 162.92 | - |

## 3. Field Experiment of the Path Planning Techniques on the Fish Farm

This work is intended to validate the conclusions derived from our simulations by implementing these algorithms onto drone hardware. We apply the three methods to pond coordinates in the Southern Illinois University Aquaculture Research Center and returns a set of routes for each method. These routes are then loaded onto a set of three drones, and power measurements from the drones are recorded after they fly their missions to conduct their measurements.

### 3.1. Experiment Setup

The drones used to facilitate this study were the SwellPro SplashDrone 4, which are designed and manufactured in Shenzhen, China. They are IP67 waterproof drones meant for hobbyist photography and aquatic activities which include waterproof connectors for payload, camera and gimbal controls. Three drones were used to conduct our experiment. Attached to the underside of the drone is a housing for communication equipment and the winch release mechanism, see Figure 5. The winch holds the sensor payload, which is released into the pond upon landing on the surface of the water down to a depth of 2 m. Therefore, the system is capable of measuring the vertical DO distributions in the pond. The payload consists of a Seeed Studio XIAO nRF52840 Sense board which is used to collect DO, temperature and pressure data and send it over Bluetooth Low Energy wirelessly to an ESP32 Wi-Fi LoRa V2 board contained in the winch mechanism, where it is then forwarded

over LoRa to an identical ESP32 board functioning as the base station. The base station receives the sensor data and uploads it to a cloud Firebase instance to automatically update an android app with the water quality status of each pond, see Figure 6 for the android app interface. Figure 7 describes the full system diagram.

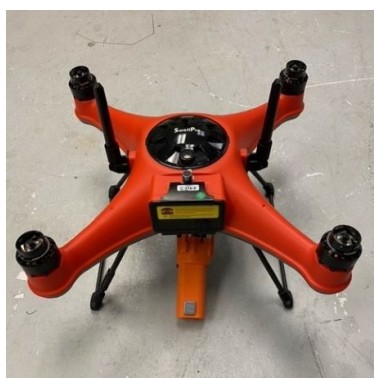
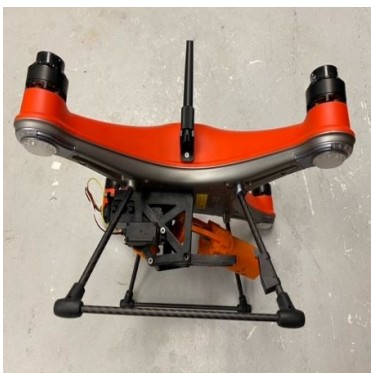
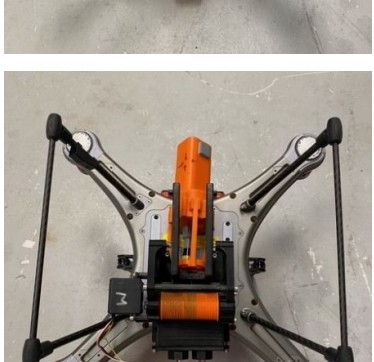
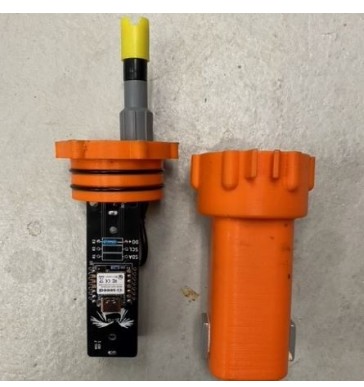

**Figure 5.** (**Top Left**) SwellPro SplashDrone 4 with orange payload visible underneath. (**Top Right**) Side view with the winch release mechanism and radio housing in black. (**Bottom Left**) Underside with the orange winch drum visible. (**Bottom Right**) Payload which contains BLE transmitter and DO, temperature and pressure sensors.

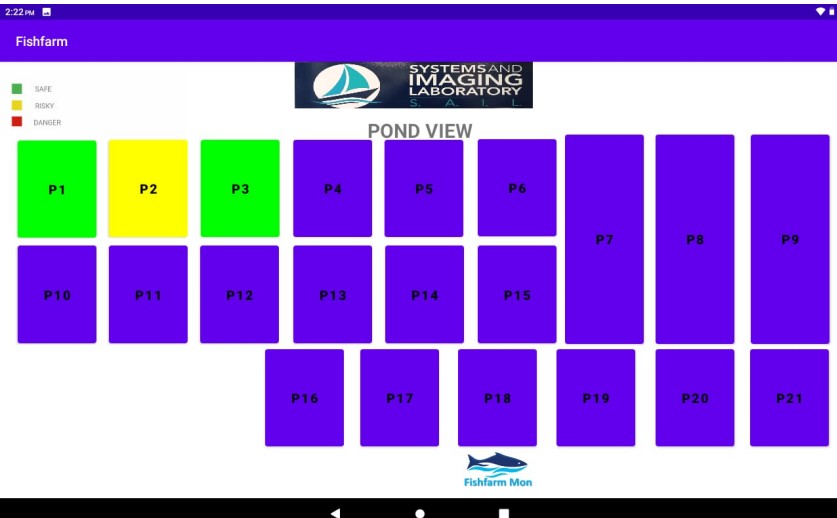

**Figure 6.** Android app displaying water quality readings sent by the drone.

The SwellPro Software Development Kit (SDK) was used as the interface between the path planning methods and the flight controllers. The custom flight control software was developed in Python to load these routes into the drone and facilitate the communication between the drones and the base station. A laptop was connected to each of the drones'

handheld controllers via dual band 802.11 Wi-Fi, switching connections as necessary to send commands to each of the drones. Mission commands with the planned route coordinates and necessary takeoff and landing commands were sent over TCP/IP to each controller and were then forwarded to the respective drone over a proprietary SwellPro radio link. Once the mission plan was successfully received by the drone, no further radio communication was necessary for flight control, and they could fly out of radio range. The LoRa link to the base station is employed to upload the pond data once it was collected.

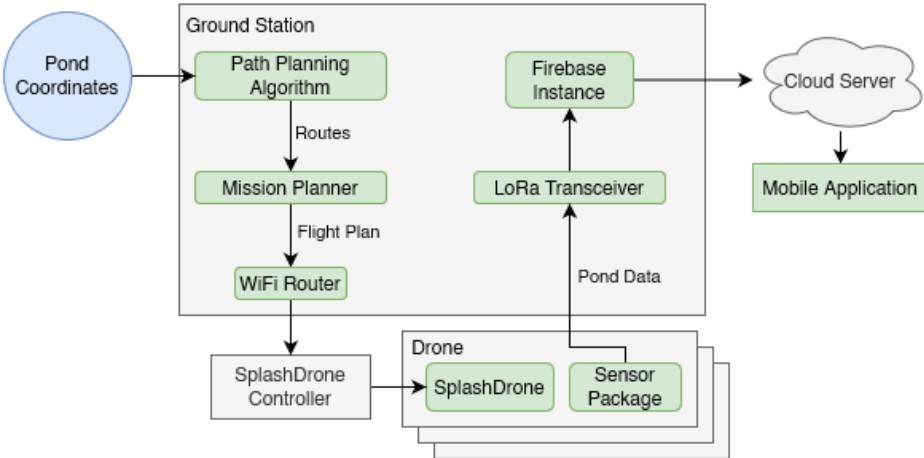

**Figure 7.** System diagram of the HAUCS path planning software and drone hardware.

### 3.2. Control Software Development

To produce the mission plan, the custom flight control software takes in a text file of the list of pond coordinates produced by the three routing techniques and generates a set of waypoints with appropriate landing, takeoff, and payload activation commands to successfully and reliably land in the water, activate the sensor payload to lower and return, take off and report the data to the base station. The payload was activated through the camera activation pin in the SplashDrone momentarily before it lands in the water, whereupon receiving the signal would lower the winch and begin sensing, then after a set time would retract the winch and stop sensing. The payload was then activated again once the drone reaches its maximum altitude to send the data to the base station. The software used to transmit missions to the SplashDrones, along with the GLOP and HPP methods, are available at https://github.com/tonydavis629/HAUCS, accessed on 6 February 2023.

A satellite image of the Southern Illinois University Aquaculture Research Center is seen in Figure 8. This facility has the capacity for 90 ponds, however at the time of testing only 26 ponds were safe to land in. Issues such as lack of water, excessive overgrowth, and power line obstacles prevented the use of 64 ponds. These 26 available ponds were loaded into the three path planning methods: GLOP, GAM and HPP.

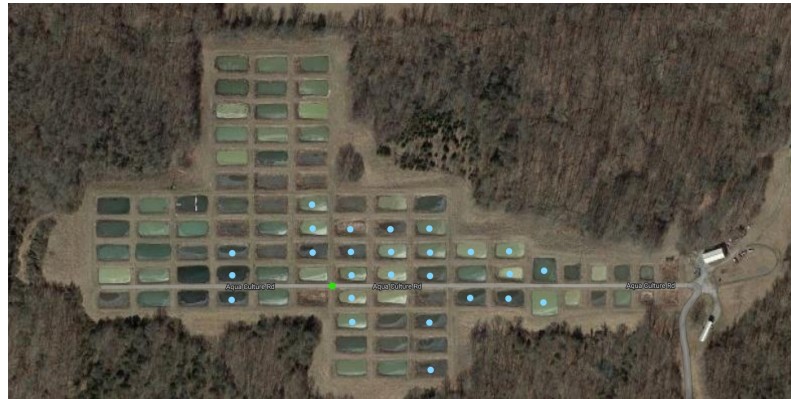

**Figure 8.** Southern Illinois University Aquaculture Research Center. The takeoff location is marked with a green dot, all available ponds are marked with a blue dot.

## 4. Results

The GPS coordinates of each of the ponds were recorded and normalized between 0 and 1 to facilitate the computation of the routes. These normalized list of coordinates were input into each of the three path planning methods, the output of each being three lists of the GPS coordinates of each pond to be visited by each drone. These three routes were then loaded into the drones using our mission control software for flight testing. During the flight, GPS, compass, battery, and temperature data were recorded. The routes for each method are simulated and visualized in Figure 9.

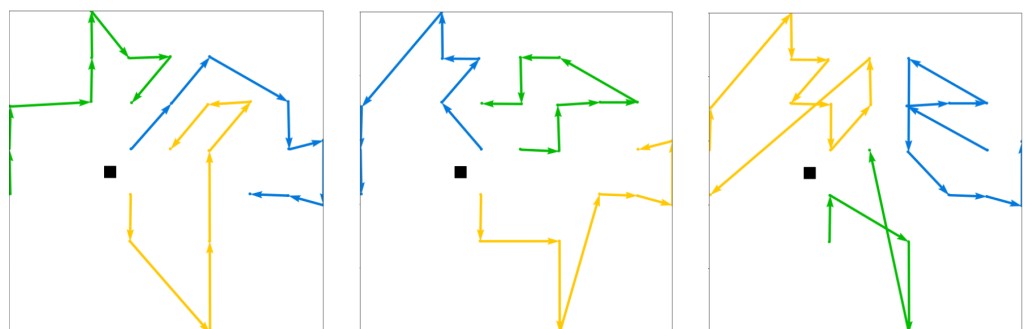

**Figure 9.** Routes for each path planning method, from left to right: GLOP, GAM, HPP. The black square signifies the launch point. Plotted with tools from [34].

### 4.1. Simulated Data

For each path planning method, the simulated distance is calculated as a means to estimate the relative energy draw and time required from each drone, see Table 7. As expected by the results of our previous study, GLOP and GAM are comparatively efficient and HPP produces significantly longer routes. HPP works on the assumption that the ponds are evenly distributed, which was not valid for our tests because only a small number of ponds on the farm were available during our tests. As a result, HPP is not effective in this situation. If all 90 ponds were available, it would be expected to be comparatively optimal to the others, while also allowing for better airspace deconfliction and requiring minimal computation.

**Table 7.** Simulated SIU Route Cost.

| Method | Drone 1 | Drone 2 | Drone 3 | Total | Average |
|---|---|---|---|---|---|
| GLOP | 1.91 | 1.87 | 1.91 | 5.70 | 1.90 |
| GAM | 2.35 | 1.64 | 1.73 | 5.72 | 1.91 |
| HPP | 2.9 | 1.73 | 3.01 | 7.64 | 2.55 |

### 4.2. Flight Data

Power data from the flights were recorded using the SplashDrone's flight log in order to validate the simulated power draw of each method. The SplashDrone flight logger is a proprietary SwellPro tool which saves data on the same device as the battery and was downloaded after the completion of all automated test flights. DO, temperature, and pressure data were not recorded as they were not the focus of this path planning experiment. Tables 8–10 shows similar results to the simulated data. GLOP is slightly more optimal than GAM, which are both significantly more efficient than HPP. In terms of power draw, GLOP and GAM were not significantly different. These results give validation to the assumption that using calculated horizontal distance is a good proxy for power consumption, and our simulations can give a good approximation of the range requirements for drones in HAUCS. This assumption may not hold for situations such as high winds or farms with significant changes in elevation.

**Table 8.** Flight Time (seconds).

| Method | Drone 1 | Drone 2 | Drone 3 | Total | Average |
|--------|---------|---------|---------|-------|---------|
| GLOP | 586 | 621 | 448 | 1655 | 552 |
| GAM | 689 | 517 | 586 | 1792 | 597 |
| HPP | 863 | 426 | 862 | 2151 | 717 |

**Table 9.** Battery Drain (mAh).

| Method | Drone 1 | Drone 2 | Drone 3 | Total | Average |
|--------|---------|---------|---------|-------|---------|
| GLOP | 1518 | 1716 | 1848 | 5082 | 1694 |
| GAM | 1914 | 1584 | 1650 | 5148 | 1716 |
| HPP | 2442 | 1254 | 2442 | 6138 | 2046 |

**Table 10.** Battery Drain (%).

| Method | Drone 1 | Drone 2 | Drone 3 | Total | Average |
|--------|---------|---------|---------|-------|---------|
| GLOP | 23 | 26 | 28 | 77 | 26 |
| GAM | 29 | 24 | 25 | 78 | 26 |
| HPP | 37 | 19 | 37 | 93 | 31 |

Wind conditions on the day of our testing were 0–5 mph in variable directions, so we did not have the opportunity to validate our simulation of steady wind conditions on path planning efficiency.

### 4.3. Issues

A number of hardware and software failures were experienced during the automated flight tests. Because the SplashDrone does not have an absolute above-sea-level altitude estimate, it is designed to reach a desired altitude relative to the takeoff location. This causes issues when the elevation of the ponds are significantly different because the drone will reach inconsistent altitudes when traveling between the ponds. This is not only a problem for airspace deconfliction, but it requires that the drone's altitude be set according to the maximum difference in elevation of its ponds. This is not ideal from a battery or safety perspective, and supports the use of a flight controller which can be programmed to take off to a specified sea level altitude, or the use of an altitude tracking device such as a downward facing lidar. This way, the drones can be assigned to certain altitudes to avoid the possibility of in-air collisions. Using an above-ground-level altitude assigned may also work and reduce the amount of altitude required to keep airspace deconflicted, but introduces greater complexity as the drones would be changing altitude between ponds.

Toward the end of our testing, one of the drones experienced a water leak during a pond landing and was subsequently damaged. This is a critical concern for HAUCS, as the drones will be expected to work robustly without leaks. This particular drone had crashed earlier during testing, so the damage from that crash likely caused a fault in a seal or a crack in the body. Future iterations of HAUCS will require great care in the design of the drone body to ensure the testability of a watertight seal in the event of a crash and as a matter of regular maintenance. It is worth mentioning that the HAUCS payload design is platform-neutral and can be easily integrated into other drones.

Another failure occurred which resulted in the drone directing itself off course and into the tree line, which was likely caused by a glitch of the onboard GPS. Though high accuracy GPS is not required, the GPS-based navigation needs to be reliable and fault-tolerant, especially due to the risk of crashing during automated flights. Better design of the flight controller may be required to address such an issue. The SplashDrone did not provide much useful data in flight logs, making certain issues difficult to troubleshoot. Object avoidance measures such as radar or vision would also be a possible solution, as supported in the open-source flight control software Ardupilot.

## 5. Discussion

These initial results validate the use of drones as a means to monitor water quality levels in an aquaculture environment. With simple software and off-the-shelf hardware, precalculated missions can be sent to drones, which dutifully execute and send data to a cloud service for real-time monitoring. This system has the potential to reduce the cost of farmed fish and address the labor shortage in the North American aquaculture industry.

Based on our simulations, GLOP and GAM path planning methods were more efficient in creating drone routes for smaller farms (generally less than 200 ponds) and HPP was more efficient in larger farms, with the advantage becoming most evident in the largest simulated farms. This principle seemed to hold during our experiments on a 26 pond farm, as in Figure 10. As was expected, GLOP and GAM performed nearly identically in our flight testing. The HPP routing method was only successful on larger simulated farms due to its simple heuristic, which requires an even spread of pond locations to visit. As a result, HPP generally performed 15–20% less efficiently at the SIU Aquaculture Research Center ponds. Additional testing will also be necessary to further validate the three path planning methods. Testing HPP on a large farm could validate its use as a large-scale path planning algorithm for aquaculture monitoring. Most commercial farms are indeed much larger than the SIU facilities, so its validation would be highly valuable.

Extrapolating from the data collected for these 26 ponds, estimations for drone range capacity requirements can now be made for other farms. Considering that measuring 9 ponds took approximately 26% battery capacity, all 90 ponds of the SIU Aquaculture Research Center could reliably be measured by 3–5 SplashDrones. Additional data will need to be collected from other facilities to give a better picture of the range requirements in general, as different farms will have different-sized ponds with different distances between them.

Significant work is still required before HAUCS is ready to support automated farm monitoring operations. The reliability and robustness of automated flights are of the utmost importance, and as demonstrated by our system failures, it is still a work in progress. Additional testing is required to improve the robustness of GPS-guided flight in pond aquaculture setting. Lightweight object avoidance systems for UAVs do exist [43,44], and may need to be implemented for safety and reliability.

Inclement conditions such as wind and rain provide an additional complication for HAUCS, as fish ponds need to be monitored particularly closely in these conditions. High winds and precipitation can have unpredictable effects on DO levels because of the stratification of oxygen in the ponds. Combined with the fact that flights may be inhibited or grounded, inclement weather is an important factor to consider in the design of HAUCS. High winds may make certain routes more inefficient if they involve traveling upwind

unnecessarily. Path plans may also need to be reconstructed on the fly due to changes in conditions or hardware failures. Real-time monitoring and flight adjustment will be required of HAUCS to operate in all conditions, as well as proper user notification in the event that flights cannot be conducted. An inclement weather protocol may also involve the use of protective waypoints for drones to take shelter temporarily without the need to return to the home station.

Future work on this project may involve using time series prediction algorithms to identify ponds which are most at risk to prioritize them for measurement. This will ease some of the burden of measuring all ponds, focusing only on those most struggling to maintain appropriate DO levels. Additionally, if vision capabilities are added to the HAUCS drones, it may be valuable to consider a mesh network between drones to enable higher bandwidth communication with the base station, as the LoRa links provide relatively low bandwidth.

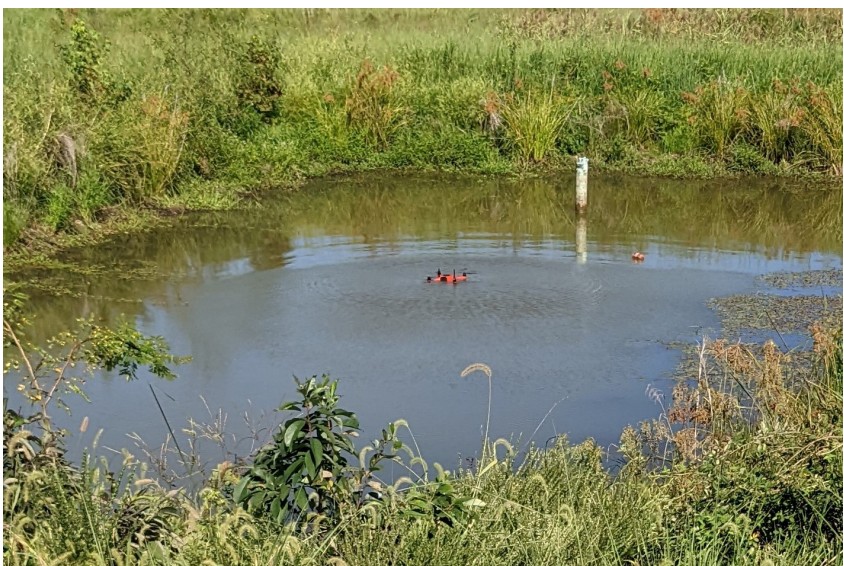

**Figure 10.** SplashDrone landing in a pond.

## 6. Conclusions

The Hybrid Aerial and Underwater Robotic System is a promising direction for reducing labor costs in aquaculture farms. The concept has been experimentally validated for its potential, but it requires significant development for the technology to be fully mature. Additional testing must be completed on larger farms to confirm the expected performance of the HPP method, as well as to collect data on varied aquaculture environments. Drones which are equipped with accurate altitude instruments, robust waterproofing, and some means of object avoidance should be used in future tests to address problems which were identified in this study. Flights in windy and rainy conditions pose an especially difficult problem due to the elevated need for water measurements and the danger of flying in inclement weather.

**Author Contributions:** Conceptualization, P.S.W., J.E.G. and B.O.; Data curation, P.S.W.; Funding acquisition, B.O.; Investigation, A.D., W.F. and M.A.K.; Methodology, W.F.; Software, A.D., W.F. and M.A.K.; Supervision, P.S.W.; Writing—original draft, A.D.; Writing—review and editing, A.D. and B.O. All authors have read and agreed to the published version of the manuscript.

**Funding:** This work was supported in part by the U.S. Department of Agriculture, National Institute of Food and Agriculture through the National Robotics Initiative 2.0: Ubiquitous Collaborative Robots program (NRI-2.0), under Grant 2019-67022-29204, Aquaculture Specialty License Plate funds granted through the Harbor Branch Oceanographic Institute Foundation and Harbor Branch Oceanographic Institute Foundation Summer Internship.

**Data Availability Statement:** Collected data can be found at https://github.com/tonydavis629/HAUCS accessed on 6 February 2023.

**Acknowledgments:** The authors would like to acknowledge the students and staff of SIU for the their assistance during our testing.

**Conflicts of Interest:** The authors declare no conflict of interest.

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
