# Peer review of "Developing and Field Testing Path Planning for Robotic Aquaculture Water Quality Monitoring"

_applsci, doi:10.3390/app13052805_

Round 1

Reviewer 1 Report

The paper ‘Developing and Field Testing Path Planning for Robotic Aquaculture Water Quality Monitoring’ discusses the design and the performances of path planning algorithms for unmanned aerial vehicles (UAV) or drones integrated with underwater measurement devices to collect the in-situ water quality monitoring data from aquaculture ponds forming a Hybrid Aerial Underwater Robotic System (HAUCS) framework.

Three methods of path-panning for the UAVs are tested, a Graph Attention Model (GAM), the Google Linear Optimization Package (GLOP), and the proposed HAUCS Path Planning algorithm (HPP). Simulated and experimental results are presented.

 The paper theme is interesting and well-written and the methodology seems to be promising. Anyway, there is an important aspect that the authors should clarify about the specific innovative algorithm that they introduce (HPP): the simulated and experimental results (section Results) demonstrate that the proposed algorithm is less performing than the other know algorithms (GLOM and GLAM). Why should this paper be published showing that the results of the proposed algorithms are worse than other known algorithms?

More some more aspects should be clarified:

-          What are the units of measure for the distances in Tables 1 and 2?

-          The better performance of the proposed algorithm HPP is evident (Tables 1-3) only for a number of ponds superior to 300.  Do you think this is a frequent situation in real farms? Please specify it.

Author Response

Dear Reviewer,

Thank you for your feedback, allow me to address your concerns.

Regarding your first point “Why should this paper be published showing that the results of the proposed algorithms are worse than other known algorithms?” The simulated data shows that the HPP method outperforms the other known methods for large farms (greater than 300 ponds). We unfortunately did not have the opportunity to experimentally validate the method on large farms because of the limited space at our testing facility. I do not believe that should preclude the paper from publication, future work can validate the HPP method at larger scales.

There are no units for Table 1 and Table 2 (as well as 4,5 and 7) because it is a simulated distance. I avoided specifying a unit to avoid the confusion that the drones are capable of such distances, for example, labelling the units as km would be inaccurate. Each simulated pond is created in a location between 0 and 1 in the x and y, so the distance unit is somewhat meaningless. I agree how this be confusing. Do you have a suggestion how to label such a simulated distance unit?

Regarding the point of HPP performance on larger farms. Commercial aquaculture farms are frequently larger than 300 ponds, and may have up to 1000 ponds. I have noted in line 80, “While most studies focus on randomly generated node locations with 100 nodes or fewer, the reality of aquaculture farms is they may have hundreds of ponds.” I will emphasize the proportion of large commercial aquaculture farms to smaller farms.

Reviewer 2 Report

Dear authors.

I have read the article carefully, I see that it has good issues for the studies of connecting IoT systems and Drones in aquaculture. I only have several comments like as.

1. The manuscript describes the Hybrid Aerial and Underwater Robotic System for water quality monitoring is not clear, you should represent this relationship by figure or diagram for better understanding.

2. In the Tables 1-3, development of an effective path planning algorithm to be able to sample all the ponds on the farm with minimal resources should be presented in formula form. Please, how many drones simulations is a swarm. In experiments, authors gave 3 drones, how to determine this number of drones.

3. In Table 1-2, what is the unit of measure for the values in the table. Authors should explain more of these tables, so that readers can understand more easily.

4. The authors should also provide simulation images of the scale of the simulated farms.

5. In section of wind considerations, authors should present this optimization in the form of a flowchart, so that the content is more effective.

6. Field experiments of the path planning on the aquaculture farm was conducted on 3 drones, the authors should also briefly talk about how collecting their energy consumptions.

7. Experimentation needs to be quantified and discussed further in the discussion of the results obtained.

The paper is a good technical issue for the use of UAVs in monitoring water parameters in ponds. I hope the authors should revise to be published.

Best regards.

Author Response

Dear reviewer,

Thank you for your feedback, allow me to address your concerns.

  1. That is a good suggestion, I will add a diagram of the HAUCS system.
  2. The number of drones used in our experiments was based on the availability of our hardware, not a formula. The simulations used 5 drones, while our experiments use 3. The end user of the system will have to determine how many drones they need based on the drone capabilities and monitoring needs. I can emphasize that, but there is no exact formula for how many drones will be required.
  3. The simulated distance measurements are unitless. They do not represent a measure of distance in the real world, and are therefore unlabeled. The unitless distances are only meaningful as a comparison in performance, not as a measure of a drone's actual range or speed. Do you have a suggestion on how to make this clear?
  4. I am uncertain if I understand this suggestion. Figure 2 showcases an example of a simulated farm. Do you recommend more such images?
  5. There is no optimization discussed regarding wind conditions. The only modification to the simulation in the wind condition is that the cost of travel is higher in the wind direction. I do not believe a flowchart would help communicate this because there is only one change made to the simulation. Can you tell me what is unclear about this change to add in a wind factor, maybe I can explain it better through the text?
  6. Line 264 reads, "Power data from the flights were recorded using the SplashDrone’s flight log in order to validate the simulated power draw of each method." I can some further details here.
  7. I will summarize more of the quantifiable results in the discussion section.

Reviewer 3 Report

In this article, a method of collecting measurements of different ponds on the farm by a set of GPS navigated drones with sensors and radio modules is presented. The paper is interesting but the path planning algorithms are not presented at all, only mentioned. The paper definitely needs to be refined, as the methods of path planning are not presented. Referring to the literature is not enough, because the path planning methods are applied to different problems. The paper should be strongly improved taking into account the following comments:

Very important problem is the use of the phrase “the most important” many times in the paper. “The most optimal” does not exist in the optimisation problems. Please use “optimal” or “close to optimal” instead of “the most optimal”. 

Instead of listing three methods: “Google Linear Optimization Package”, “Graph Attention Model” or “the HAUCS Path Planning Algorithm”, please present at least one with all details and a numerical example. Now the paper does not describe the method adequately.

Author Response

Dear reviewer,

We can certainly add more details explaining our methods to make it more clear. The numerical example I think would best be illustrated as a figure, showing how routes are calculated for each step.

I have adjusted some of the language to reflect your recommendations.

Round 2

Reviewer 1 Report

Ok, let future work validate the HPP method at larger scale.  I don't have any suggestion for  labelling simulated distance units, a clear explanation could be enough. 

Reviewer 3 Report

The disadvantage of the paper is listing three methods: “Google Linear Optimization Package”, “Graph Attention Model” or “the HAUCS Path Planning Algorithm”, without description, the advantages are the application of the methods and presented results. Hovewer, the authors refer the literature where the methods can be found. Therefore, I recommend accepting the paper in its current form.